# Intravascular Lymphoma: A Unique Pattern Underlying a Protean Disease

**DOI:** 10.3390/cancers17142355

**Published:** 2025-07-15

**Authors:** Mario Della Mura, Joana Sorino, Filippo Emanuele Angiuli, Gerardo Cazzato, Francesco Gaudio, Giuseppe Ingravallo

**Affiliations:** 1Section of Molecular Pathology, Department of Precision and Regenerative Medicine and Ionian Area (DiMePRe-J), University of Bari “Aldo Moro”, 70124 Bari, Italy; m.dellamura1@studenti.uniba.it (M.D.M.); giuseppe.ingravallo@uniba.it (G.I.); 2Unit of Haematology, Department of Medicine and Surgeon, F. Miulli University Hospital, LUM University, Casamassima, 70010 Bari, Italy; fragaudio@alice.it

**Keywords:** intravascular lymphoma, intravascular lymphomatosis, non-Hodgkin lymphoma, B-cell lymphoma, NK/T-cell lymphoma

## Abstract

Intravascular large B-cell lymphoma (IVLBCL) is a rare and aggressive subtype of diffuse large B-cell lymphoma characterized by the selective intravascular growth of neoplastic lymphoid cells, typically without the formation of solid tumor masses or lymphadenopathy. This unusual presentation often leads to delayed or missed diagnoses, frequently confirmed only post-mortem. In this study, we provide a detailed clinicopathological analysis of IVLBCL, highlighting its histological features, immunophenotypic profile, molecular characteristics, and potential mechanisms of immune evasion, including PD-L1 expression and HLA downregulation. We also discuss the current diagnostic challenges, emphasizing the importance of a multidisciplinary approach, and outline the possible therapeutic strategies to improve outcomes in patients affected by this aggressive malignancy.

## 1. Introduction

Intravascular lymphoma (IVL) is a rare and aggressive form of non-Hodgkin’s lymphoma (NHL) characterized by the selective growth of neoplastic cells within the lumina of blood vessels. In most cases, the neoplastic elements exhibit a B-cell phenotype—known as intravascular large B-cell lymphoma (IVLBCL)—although sporadic cases with a NK/T-cell phenotype—termed intravascular NK/T-cell lymphoma (IVNKTCL)—have also been reported.

The disease was first described in 1959 by Pfleger, who believed it was a neoplasm of endothelial origin and referred to is as ‘systematized endotheliomatosis’. In 1986, Sheibani demonstrated it is made up of lymphocytes, so it was relabeled ‘angiotropic large cell lymphoma’. Terms such as ‘malignant angioendotheliomatosis’, ‘angioendotheliotropic lymphoma’, and ‘intravascular lymphomatosis’ had also been employed.

Currently, IVLBCL is recognized as a distinct clinicopathologic entity by the WHO Classification of Haematolymphoid Tumours (5th edition) and is defined as an aggressive extranodal B-cell lymphoma (BCL) in which large neoplastic proliferate virtually exclusively within the lumina of blood vessels [1].

In contrast, IVKNTCLs do not constitute a disease-defining group, and there is ongoing debate as to whether they are more closely related to aggressive NK leukemia (ANKL) or to extranodal NK/T-cell lymphoma, nasal type (ENKTL).

Most of the available literature on IVL is based on single case reports and small case series, making it difficult to draw definite conclusions about clinical and biological properties of the disease. Herein, we provide a comprehensive review of intravascular lymphomas, dividing the paper into three sections. The first section is dedicated to IVLBCL, by far the most frequent subtype, with a focus on its clinical and morphological features, pathogenetic mechanisms, and therapeutic perspectives. The second section addresses the rare cases of IVNKTCL. Finally, the third section discusses systemic lymphomas with angiotropic/angiocentric growth or an intravascular component, which must always be considered and excluded before a definitive diagnosis of intravascular lymphoma can be established.

## 2. IVLBCL: The Archetype of Intravascular Lymphomas

### 2.1. Epidemiology

IVLBCL is a rare neoplasm mostly affecting elderly individuals, with an incidence below 1/1.000.000 person/year, a median age at diagnosis of 67 years (range: 23–92) and an approximately equal male-to-female ratio [2,3].

A prior medical history of hematopoietic tumors, non-hematopoietic tumors, and autoimmune disorders is found in 12%, 20%, and 11% of patients, respectively; thus, the incidence of hematopoietic malignancies appears to be significantly higher than that observed in the general population [4]. Additionally, cases have been reported in the context of HIV/AIDS and in patients undergoing active immunosuppressive therapy [5].

Among hematopoietic tumors, NHLs are frequently associated. They can occur before, concomitantly, or after IVLBCL diagnosis and are mostly non-clonally related. Occasionally, a clonal relationship has been demonstrated: in such instances, IVLBCL represented the recurrence of a DLBCL [6] or a rare example of Richter transformation [7,8].

Among non-hematopoietic tumors, both benign and malignant epithelial or mesenchymal neoplasms may coexist. In such collision cases, IVLBCL is incidentally identified on histopathologic examination, like due to its ‘tumor-tropism’ (i.e., the tendency to grow within solid tumor-associated vessels) [9].

Among autoimmune diseases, systemic lupus erythematosus (LES), rheumatoid arthritis, vasculitis, Hashimoto’s thyroiditis, Sjogren’s syndrome, and celiac disease have been reported [4].

### 2.2. Clinical, Laboratory, and Imaging Features

#### 2.2.1. Clinical Presentation

Clinical manifestations of IVLBCL are extremely variable: in fact, it can virtually involve all body districts due to its disseminated nature, provoking organ dysfunction through the occlusion of blood vessels.

Three main patterns of clinical presentation can be observed: the classic variant, the hemophagocytic syndrome-associated variant (HPS-associated variant), and the cutaneous variant (Figure 1).

The **classic variant** is characterized by both visceral and cutaneous involvement. It is also called the Western variant, as it is the most common type of presentation in Western countries. Central nervous system (CNS) involvement is frequent and presents with diffuse, more rarely focal, neurological signs including altered consciousness, cognitive decline, vertigo, amaurosis, aphasia, dysarthria, hyposthenia, sensorimotor deficits, and seizures. Neurolymphomatosis may also occur. These neurological manifestations usually fully regress after treatment [10]. Moreover, respiratory failure and endocrine disfunction (i.e., hypopituitarism, hypothyroidism, hypoadrenalism) are frequently observed [11,12], likely due to the high vascularization of these organs, which may facilitate intravascular lymphoid cell proliferation and consequent functional impairment. Cutaneous involvement displays a broad clinical spectrum, including solitary or multiple erythematous or violaceous nodules and plaques as well as diffuse eruptions consisting of erythematous-desquamative or papular lesions. They commonly localize to the mammary, abdominal, or extremity regions. Fever of unknown origin (FUO) and other B symptoms (unintentional weight loss and night sweats) are present in over half of patients. A rapid decline in performance status is almost invariably observed [11].

The **HPS-associated variant** is characterized by multi-organ failure, hepatosplenomegaly, and pancytopenia attributable to concurrent HPS; rapid onset and progression, fever and B-symptoms are constant features. It is also called the Asian variant, because it is the most common type of presentation in Eastern countries [11]. Currently, HPS is defined by meeting at least five of the following nine diagnostic criteria: (I) fever ≥ 38.5 °C; (II) splenomegaly; (III) cytopenia in almost two hematopoietic lineages (Hb < 9 g/dL, platelets < 100 × 103/μL, absolute neutrophil count < 100/μL); (IV) hypertriglyceridemia (≥3.0 mmol/L); (V) hypofibrinogenemia (≤150 mg/dL); (VI) hyperferritinemia (≥500 μg/L); (VII) elevated soluble levels of IL-2Rα/CD25 (>2400 U/mL); (VIII) histopathological evidence of hemophagocytosis in bone marrow, spleen, or lymph nodes biopsies; and (IX) reduced or absent NK cell activity [13,14]. To facilitate early clinical recognition, the combination of fever, splenomegaly, thrombocytopenia, and hyperferritinemia has been also proposed as a minimal clinical set for HPS diagnosis [13,15].

The **cutaneous variant** is characterized by isolated skin involvement, with no evidence of systemic disease at staging. It essentially occurs in Western female patients and is associated with a favorable prognosis. Constitutional symptoms are rarely observed, until the disease remains confined to the skin [1,11].

Currently, the biological differences between classic and HPS-associated variants remain unclear, although it is hypothesized that genetic and epigenetic patient-specific factors may influence disease presentation. The cutaneous variant may represent an early, localized phase of the disease and shows an invariable tendency to progress without treatment [12,16].

#### 2.2.2. Laboratory Findings

Common laboratory abnormalities include anemia (63–69%), thrombocytopenia (29–58%), and leukopenia (24–27%) as well as elevation in lactate dehydrogenase (LDH; >95%), sIL-2Rα/CD25, ferritin, C-reactive protein (CPR), and erythrocyte sedimentation rate (ESR). Hypoalbuminemia, presence of a monoclonal serum component, and alterations in organ function parameters can be observed, with the latter considered surrogates of end-organ involvement [2,10,11,16]. Flow cytometry and peripheral blood (PB) smears are rarely positive (<25%), reflecting the limited tendency from neoplastic cells to recirculate. Occasionally, a “leukemic” presentation has been described, characterized by ≥10% circulating neoplastic lymphoid cells among white blood cells, which correlates with an unfavorable prognosis [17]. Cerebrospinal fluid (CSF) analysis is usually non-diagnostic, even in the presence of CNS involvement [7], typically revealing only hyperproteinorrachia or mild reactive pleocytosis [3].

#### 2.2.3. Imaging

Due to the lack of tumor mass formation or lymphadenopathy, IVLBCL may be associated with a substantial tumor burden despite unremarkable radiological findings [12].

**Fluorodeoxyglucose positron emission tomography (FDG-PET)** is the most sensitive imaging modality for detecting disease localizations, particularly in the lungs and bone marrow (BM), even in the absence of overt clinical or instrumental abnormalities. However, false-negative PET findings may occur, primarily due to low tumor cell density or reduced metabolic activity [10,11,18].

Central nervous system (CNS) involvement is best assessed using **magnetic resonance imaging (MRI)**, which typically reveals one or more of five nonspecific patterns—usually in form of multifocal lesions—that generally regress with therapy: (1) infarct-like lesions appearing as persistent hyperintense areas on diffusion-weighted imaging (DWI), with or without reduced apparent diffusion coefficient (ADC) values; (2) nonspecific white matter lesions seen as hyperintensities on T2-weighted or FLAIR sequences and sub-classified as small (<3 mm in maximum diameter) or large (>3 mm); (3) leptomeningeal lesions defined as spontaneous (hemorrhage-unrelated) leptomeningeal hyperintensity on T2/FLAIR or as enhancement following gadolinium administration; (4) mass-like lesions; and (5) hemorrhagic lesions observed as hypointense areas on T2* or susceptibility-weighted imaging (SWI). Recognition of these findings may aid in early diagnosis and accurate disease staging [3,19].

Moreover, pulmonary involvement may be suggested on computed tomography (CT) by the presence of ground-glass opacities, nodules, and pleural effusion [7,10,11].

### 2.3. Diagnosis

IVLBCL represents a significant diagnostic challenge given its rarity, polymorphic clinical presentation, and lack of pathognomonic laboratory or imaging findings. All these factors contribute to delayed diagnosis and, consequently, therapeutic intervention, which is associated with poorer clinical outcomes [2,12].

The current diagnostic gold standard relies on histological demonstration of neoplastic intravascular lymphoid cells in tissue specimens. Every time clinical suspicion arises, both **skin biopsy** and **bone marrow biopsy** (BMB) should be promptly performed (Figure 2).

Clinical elements suggesting IVLBCL include rapidly progressive illness with marked deterioration in general condition, absence of detectable tumor masses or lymphadenopathy, anemia, and elevation in LDH and ferritin, particularly in the presence of concurrent neurological or cutaneous signs.

Then, when skin lesions are present, a target biopsy should be performed. In the absence of visible lesions, random skin biopsies (RSBs) should be executed as they frequently reveal microscopic evidence of the disease, especially within subcutaneous blood vessels. RSBs can be performed as incisional or telescoping biopsies. They should include subcutaneous tissue to a depth of at least 5 mm; thus, punch biopsies are inadequate for diagnostic purposes. The recommended protocol involves obtaining one sample from 3–4 different anatomical sites, such as the thighs, posterior upper arms, and abdomen. It may also be useful to sample benign hemangiomas, if present, given their potential colonization by lymphomatous cells.

The addition of a BMB provides both increased diagnostic sensitivity—by identifying cases lacking cutaneous involvement—and valuable information for disease staging. Furthermore, the recognition of IVLBCL is often associated with complex karyotypes and multiple recurrent cytogenetic abnormalities, which can be useful in suggesting the presence of occult disease even in morphologically negative bone marrow.

To avoid false-negative results, it is advisable to delay biopsies for 3–5 days following the last dose of corticosteroids.

This combined approach enables the identification of neoplastic cells in tissue sections in almost all cases, while avoiding the invasiveness and morbidity associated with organ-targeted biopsies; however, in instances where histological confirmation is not achieved and clinical suspicion remains high, random gastrointestinal or transbronchial lung biopsies or organ-targeted biopsies are warranted [11,20,21].

Not rarely, diagnosis is achieved only at autopsy, especially in patients with rapid clinical deterioration and when suspicion has not been raised by clinicians. In such cases, despite frequently unremarkable gross findings, extensive sampling from multiple organs is essential to identify the presence of lymphoma cells.

Once the diagnosis is established, staging should be performed according to the Lugano classification, incorporating findings from BMB, imaging studies, and laboratory tests.

### 2.4. Staging

Staging is performed according to the Lugano modification of the Ann-Arbor staging system. Clinical, laboratory, and imaging findings as well as targeted biopsies can be employed for this purpose. At diagnosis, the majority of patients present with stage IV disease, most commonly due to BM involvement [11].

### 2.5. Pathology

#### 2.5.1. Histopathology

Histologically, IVLBCL consists of a proliferation of neoplastic lymphoid cells within medium and small-sized blood vessels [1] (Figure 3).

Lymphoid elements are medium-to-large in size and exhibit intermediate morphological features between centroblasts and immunoblasts/plasmablasts, showing high nuclear/cytoplasmic ratio, smooth or slight irregular nuclear contour, single prominent nucleolus or multiple nucleoli, and scant cytoplasm. Mitotic figures can be appreciated [11]. They grow within blood vessels in three main histological patterns, usually coexisting together: the **cohesive pattern**, consisting in a complete filling and obliteration of vascular lumen; the **dys-cohesive pattern**, with preferential growth in the central portion of vessels resulting in a free-floating appearance; and the **marginating pattern**, where neoplastic cells adhere to the endothelium and leave the center of the lumen relatively clear [11]. Sinusoidal involvement may be observed in the liver, spleen, and BM [1]. In particular, BM infiltration is categorized into the pure intrasinusoidal pattern, in which neoplastic cells are confined within sinusoidal spaces; the intrasinusoidal pattern with interstitial extravasation, where a portion of neoplastic cell is outside the sinusoids within the interstitium; the nodular/diffuse pattern, in which neoplastic cells diffusely proliferate into the BM [10]. The HPS-associated variant is accompanied by non-neoplastic histiocytes filled with red cells or mononuclear cells, identifiable both in PB smears and in BM samples [11].

There are some exceptions to this model.

First, although diagnosis typically requires identification of large neoplastic lymphocytes, small-sized neoplastic elements may be present in up to 18% of cases, with definitive diagnosis established via biopsy of a different tissue [22]. In rare cases, Reed–Sternberg-like cells or anaplastic morphology have also been reported [11,23]. In such cases, a careful exclusion of other lymphomas with vascular involvement is mandatory; growth exclusively limited to the lumina of blood vessels and a coherent immunophenotype can resolve the diagnostic dilemma.

Then, a certain degree of extravascular dissemination may be noted, typically consisting of isolated cells or small clusters, only occasionally forming perivascular sheets [24]. When a substantial extravascular component is present, the neoplasm should be better classified as DLBCL with intravascular involvement (DLBCL-IV) [25,26]. It is not known whether the latter represents a biological distinct entity, closer to DLBCL, or falls within the spectrum of IVLBCLs; some authors propose a closer relationship between DLBCL-IV and IVLBCL rather than classic DLBCL, based on clinical behavior and prognosis [25,27].

Moreover, rare instances of IVLBCL involving large-caliber vessels, such as aorta, truncus pulmonalis, carotids, abdominal arteries, and left ventricle, have been documented. In all of these cases, lymphoma cells formed neoplastic thrombi adherent to the endothelium, without evidence of vascular wall or perivascular tissue invasion [28,29].

Finally, tumor cells inside lymphatic vessels, highlighted by podoplanin staining, have also been noted [26].

**Differential diagnosis** of IVLBCL includes several mimickers, such as IVNKTCL, intravascular pseudolymphoma [9,30,31], intravascular dissemination of a mass-forming lymphoma (e.g., DLBCL) [9,11], and lymphomas with an angiocentric growth pattern (e.g., ENKTL) [1,32], exclusion of which is mandatory before an IVLBCL diagnosis can be established through an extensive clinical, imaging, and immunohistochemical workup. A more exhaustive discussion of these entities is provided in Section 4.

#### 2.5.2. Immunophenotype

IVLBCL cells display the immunophenotype of mature peripheral B cells, with constant expression of **pan-B-cell markers**. Strong CD20+ is observed in almost all cases; very rare CD20 cases stain positive for other B-cell markers (e.g., CD79a, PAX-5+) [11].

According to the Hans algorithm, the vast majority of IVLBCLs belong to the **non-GCB subtype** (82–87%). In fact, CD10 expression is observed in 11–13% of cases, while BCL6 expression occurs in 26–48% and MUM1 expression occurs in 64–95%, without prognostic differences [2,4,27]. These data have been confirmed by microarray-based analyses of gene expression [33].

Aberrant CD5 expression is frequently observed (22–75%) in the absence of clinical prognostic correlations [4,27]. CD3 and other T-cell markers, as well as cyclin-D1, are usually negative [4,11]. The Ki67 index ranges from 20% to 100% [2]. Occasional CD30 expression has also been reported in recurrent IVLBCL, the cutaneous variant [34].

BCL2 is positive in 82–91% and MYC in 50–60% of cases, in the absence of gene translocations or amplifications [2,27]; a double expression occurs in around 40%, correlating with an increased mortality rate [35].

Occasionally, EBV-encoded RNA (EBER) expression has been reported; its pathogenetic role as well as clinical and prognostic correlations are largely unknown [2].

A comprehensive illustration of the immunophenotype of IVLBCL is provided in Table 1.

### 2.6. Genetic Landscape

#### 2.6.1. Cytogenetics

Currently, two main studies have investigated the karyotypic profile of IVLBCL.

The first study, conducted by Klairmont et al., analyzed 29 cases. A **complex karyotype**—defined as the presence of three or more chromosomal aberrations—was identified in all of them, with a median of 10 aberrations per case. Recurrent abnormalities involved chromosomes 1 (72.4%), 3 (58.6%), 6 (58.6%), 18 (55.2%), and 14 (41.4%), with no apparent correlation with clinical presentation or outcome [21]. At least one marker chromosome was detected in 50% of patients.

**Chromosome 1 abnormalities** commonly affected the 1p13 and 1q21 regions, which harbor the *NOTCH2* and *CD56* genes, respectively. Both NOTCH2 dysregulation and CD56 loss appear to be involved in lymphomagenesis and immune evasion in DLBCLs [21].

**Chromosome 6 anomalies** typically included 6q deletions or additions of material of unknown origin. Notably, the latter can co-occur with deletions in the corresponding region, suggesting that loss of genetic material at 6q is a consistent feature. The 6q region includes the *PRDM1* tumor suppressor gene, which is frequently inactivated in non-germinal center B-cell-like (non-GCB) DLBCLs [21].

**Chromosome 14 abnormalities** primarily involved the 14q32 region, corresponding to the immunoglobulin heavy chain (IGH) gene locus. However, translocation partners such as *BCL2*, *BCL6*, *CCND1* (Cyclin D1), and *MYC* were almost never involved: currently, only one case with t [14,18] and a few cases with t [3,14] or t [11,14] have been reported [36,37,38].

**Chromosome 18 anomalies** most consisted of trisomy, a finding also observed in other BCLs, although the pathogenic significance in IVLBCL remains unclear [21].

The second study, conducted by Fujikura et al., evaluated 12 cases. Cytogenetic abnormalities were identified in six patients (50%). However, karyotyping was performed on bone marrow samples, which often contain a low tumor burden; thus, a normal karyotype may have reflected the karyotype of residual myeloid cells. In cases with detectable abnormalities, recurrent cytogenetic alterations were consistent with those reported by Klairmont, including complex karyotypes, 6q abnormalities, polysomy, and marker chromosomes. Additionally, recurrent abnormalities included chromosome 8 loss or gain of material of unknown origin on 8p, loss of chromosome 4, and addition of unknown material at 19q13. No significant clinical or prognostic differences were observed between patients with normal and abnormal karyotypes [39].

#### 2.6.2. Molecular Biology

Earlier molecular studies employing Southern blot and PCR demonstrated clonal immunoglobulin gene rearrangement and frequent occurrence of somatic hypermutation in neoplastic cells, thus confirming their origin [11].

Currently, molecular analyses can be conducted on tissue-derived DNA (tdDNA) and cell-free DNA (cfDNA, i.e., liquid biopsy). The latter has proven to be an excellent source of genetic material, with a sensitivity approaching 100%, overcoming the limitations linked to the scarcity of neoplastic cells in tissue sections. Moreover, cfDNA levels correlate with disease burden, providing valuable insights for diagnosis, molecular profiling, and response monitoring. However, its principal limitation lies in the requirement for broad sequencing approaches, due to the absence of pathognomonic mutations in IVLBCL [5,16,40].

A recent next-generation sequencing (NGS) study of tdDNA from 15 IVLBCL cases identified frequent mutations in *PIM1* (60%), *MYD88* (53%), *CD79B* (53%), *IRF4* (27%), *ETV6* (27%), *TMEM30A* (27%), and *BTG2* (27%); more rarely *NOTCH2*, *CCND3*, and *GNA13* were involved [26].

All *MYD88* mutations were the canonical p.L265P variant, while *CD79B* mutations consistently involved the immunoreceptor tyrosine-based activation motif (ITAM) domain, most frequently p.Y196. Together, ***MYD88* and *CD79B* mutations** were detected in 67% of cases and co-occurred in 40%. These mutations result in constitutive activation of Toll-like receptor and B-cell receptor signaling pathways, respectively, leading to **NF-κB pathway activation**. This, in turn, promotes anti-apoptotic signaling and uncontrolled proliferation in neoplastic cells—a mechanism well-established in various hematologic malignancies [26]. Similar mutational patterns in *MYD88* and *CD79B* have been reported by other authors in studies using both tdDNA and cfDNA [16,41,42] (Figure 4). Notably, no significant differences in clinical presentation or outcomes were observed based on mutational profiles [26,41].

A subsequent pathway enrichment analysis showed NF-kB (87%), epigenetic modifiers (47%), immune response (33%), and B-cell differentiation (27%) as the most affected ones [26]. All these data suggest IVLBCL has a genomic profile corresponding to the so-called ‘MYD88/CD79B-mutated (MCD)’ genetic subgroup of DLBCLs, a feature also shared by other primary extranodal BCLs of the ABC subtype (i.e., lymphomas of immune-privileged sites), providing a molecular background for developing novel targeted therapies [16,26,42,43].

#### 2.6.3. Immune Surveillance Evasion

Immune surveillance evasion is a critical mechanism in lymphoma biology. In particular, upregulation of immune checkpoint pathways through expression of PD-1 ligands and/or loss of major histocompatibility complex (MHC) molecules have been documented in many BCLs including IVLBCL, which can be observed in 65% of cases overall [44].

**PD-L1 and/or PD-L2 expression** has been reported in 36–44% of IVLBCL cases, almost always consisting of selective PD-L1 expression. In the literature, PD-L1 positivity has been defined by partial or complete membranous staining in at least 5–10% of neoplastic cells, as lower expression levels are rarely observed whenever PD-L1 is expressed [16,26,45,46]. In approximately 50% of PD-L1/PD-L2-positive cases, expression is associated with chromosomal alterations affecting the respective gene loci. These alterations include copy number gains and translocations, the latter often involving 3’-UTR truncations, stabilizing the transcripts and resulting in markedly increased expression of the corresponding proteins [16,44,45]. In addition, the activation of the JAK/STAT and NF-κB pathways has been implicated as an alternative mechanism driving PD-L1 upregulation [45].

PD-L1 expression is frequently observed not only in neoplastic cells but also in tumor-associated histiocytes. In rare cases, selective expression is confined to non-neoplastic components whose biological significance remains unclear [26]. No significant associations have been identified between PD-L1 expression and clinical, morphological, or immunophenotypic features [45,46]. However, PD-L1 positivity has been linked to poorer outcomes, likely due to immune evasion; in this regard, immune checkpoint inhibitor therapy has been claimed as a potential therapeutic option [45] (Figure 4).

**Loss of MHC** expression has been reported in approximately 27% of cases, predominantly involving MHC class I molecules. This alteration is more frequently observed in PD-L1-negative cases [44]. Many genetic mechanisms leading to altered MHC expression have been described in tumors, including mutations in its transcriptional transactivator (the *CIITA* gene) as well as epigenetic modifications, resulting in downregulation of MHC proteins on tumor cells. Since the only evidence of MHC downregulation in IVLBCL reported so far is based on immunohistochemical analysis [44], further investigations are warranted to elucidate the precise molecular mechanisms underlying MHC loss in IVLBCL cells.

### 2.7. Pathogenesis

The putative normal counterpart of IVLBCL is a peripheral B cell that has undergone neoplastic transformation [1], becoming “frozen” at the stage of margination and adhesion to the vascular endothelium, thereby accounting for its selective intravascular localization. The biological underpinnings of this phenomenon have been extensively investigated: in fact, blood vessels represent not only the disease vehicle but also the “home” (i.e., proliferative niche) of neoplastic lymphocytes, as demonstrated by the presence of mitotic figures and elevated Ki-67 proliferation indices.

Current evidence suggests that IVLBCL cells exhibit an **impaired ability to transmigrate across the endothelium** and **to degrade the extracellular matrix**, processes necessary for the formation of extravascular tumor masses. This deficiency, likely attributable to multiple molecular aberrations, may underlie the intravascular confinement of lymphoma cells [11,47].

Specifically, loss of CD11a/CD18 (LFA-1), a leukocyte integrin critical for firm adhesion to the endothelium prior to transendothelial migration, has been reported in IVLBCL [48]. Additionally, neoplastic cells frequently lack expression of CD29 (the β1 integrin subunit) and CD54 (ICAM-1). The first is normally present on leukocyte surface and binds to VCAM-1, an inducible endothelial counterreceptor, in a crucial mechanism for diapedesis. The second acts as a ligand for various integrins; the precise significance of its expression in neoplastic lymphocytes is controversial; however, its absence correlates with disseminated disease in NHLs [49]. A recent study confirmed these findings, demonstrating CD29-negative tumor cells in 67% and ICAM-1-negative tumor cells in 95% of IVLBCL cases, in contrast to DLBCL, where these molecules are typically expressed. The detection of rare CD29-positive and/or ICAM-1-positive cases suggests that additional mechanisms may contribute to the intravascular localization of malignant cells [4].

On the other hand, IVLBCL cells do not express matrix metalloproteinases 2 and 9 (MMP-2 and MMP-9), enzymes that facilitate tissue infiltration and parenchymal invasion [47].

Despite these insights, the mechanisms by which neoplastic B cells adhere to the endothelium and proliferate within the vascular lumen—without inducing a leukemic presentation—remain unclear. Chemokine–chemokine receptor interactions have been proposed as a potential mechanism. Case reports have described the expression of CXCL9 on lymphoma cells and its receptor CXCR3 on endothelial cells, as well as in other instances, such as CXCR4 expression on lymphoma cells and its ligand CXCL12 on the endothelium (Figure 4). A potential correlation with the pattern of organ-specific involvement and ‘tumor-tropism’ is currently unknown [50,51].

In addition, recent findings have identified a high frequency of activating mutations in *RAC2*, a member of the Rho GTPase family involved in regulating cytoskeletal dynamics and cell adhesion. *RAC2* activation promotes increased B-cell adhesion, potentially contributing to their retention within the vascular compartment. However, the limited number of cases studied thus far precludes definitive conclusions [42].

### 2.8. Therapy, Follow-Up, and Prognosis

IVLBCL is a rare and aggressive malignancy that necessitates prompt therapeutic intervention. Due to its low incidence and heterogeneous clinical presentation, standardized treatment guidelines are lacking, and therapeutic decisions rely primarily on retrospective data and clinical experience (Figure 5).

The current first-line therapy for intravascular large B-cell lymphoma (IVLBCL) is the R-CHOP regimen (rituximab, cyclophosphamide, doxorubicin, vincristine, and prednisone). Due to the limited CNS penetration of R-CHOP, CNS-directed therapy should be added, regardless of its initial involvement. ASCT may be considered in selected young patients and is associated with improved outcomes. Targeted therapies or immunotherapy can be explored as salvage options. In frail patients who are unfit for systemic treatment, the primary goal is local disease control.

The current first-line therapy for IVLBCL is the **R-CHOP regimen** (rituximab, cyclophosphamide, doxorubicin, vincristine, and prednisone). The inclusion of rituximab—supported by the consistent expression of CD20 on neoplastic B cells—has significantly improved clinical outcomes, with overall response rates exceeding 60% and a 3-year overall survival (3Y-OS) surpassing 30%, enabling durable remission in a subset of patients [4].

CNS involvement is frequent, affecting 30–40% of patients at diagnosis and emerging in another 25% during follow-up. Given the limited CNS penetration of R-CHOP, the standard of care currently includes R-CHOP in combination with **CNS-directed therapy**, regardless of initial CNS involvement [16,52]. The latter includes

intrathecal methotrexate, cytarabine, and corticosteroids for prophylaxis and treatment, offering reduced systemic toxicity and lower drug dosages—particularly beneficial in elderly or frail patients;high-dose systemic methotrexate, sometimes combined with cytarabine, serving as an alternative to the latter (in fact, these drugs can cross the blood–brain barrier);radiotherapy and spinal decompression in select cases of leptomeningeal disease.

However, other authors reported modest benefits from these CNS-directed strategies, possibly due to the low incidence of parenchymal CNS infiltration in IVLBCL. Currently, its impact on OS is still debated, and prospective data are needed [10,53].

In rare cases, local treatments such as radiotherapy for isolated cutaneous involvement or surgical resection for organ-confined disease have led to remission [5,54]. Nonetheless, these approaches are not curative and often lead to rapid progression when used in isolation. Therefore, they should be reserved for patients who are frail or have significant comorbidities.

**Autologous stem cell transplantation (ASCT)** has demonstrated encouraging outcomes in selected younger patients, with 2-year overall survival (OS) and progression-free survival (PFS) rates exceeding 80–90% in some cohorts [55]. The optimal timing of ASCT—whether as consolidation or salvage therapy—remains unclear, and its use is usually limited by patient age and health status [16].

IVLBCL shares clinical and molecular features with other high-risk subtypes of diffuse large B-cell lymphoma (DLBCL), such as double-hit and double-expressor lymphomas, and those with primary extranodal involvement (e.g., CNS, bone marrow, or peripheral blood). These aggressive variants are associated with adverse prognostic markers, including *MYC* and *BCL2* rearrangements, high proliferation rates, and immune evasion mechanisms such as PD-L1 expression. In this context, future therapeutic directions are increasingly aligned with novel strategies developed for other high-risk and relapsed/refractory aggressive BCLs:Polatuzumab vedotin, an anti-CD79b antibody–drug conjugate, has been approved in combination with R-CHOP as first-line therapy in high-risk BCLs, demonstrating superior progression-free survival compared to R-CHOP alone; it has also been employed in a refractory IVLBCL case [56];Bispecific antibodies, including glofitamab and epcoritamab (targeting CD20 and CD3), represent off-the-shelf immunotherapy options with promising efficacy and manageable toxicity profiles in relapsed/refractory settings;Chimeric antigen receptor (CAR) T-cell therapies (e.g., axicabtagene ciloleucel, tisagenlecleucel) offer potentially curative treatment for patients who have failed at least two prior lines of therapy;Immune checkpoint inhibitors, particularly those targeting the PD-1/PD-L1 axis, are under active investigation in PD-L1-positive lymphomas, aiming to circumvent immune evasion mechanisms.

Response monitoring is difficult, due to the lack of tumor masses or other easy quantifiable parameters, and essentially relies on the evaluation of clinical and laboratory indices. Prognostic tools such as the International Prognostic Index (IPI), alongside emerging biomarkers like circulating tumor DNA (ctDNA), soluble IL-2 receptor (sIL-2R), and PD-L1 expression, help stratify patient risk and guide therapy choices. Eventual relapse usually occurs within 2–3 years and correlates with a poor outcome [56].

Notably, cutaneous involvement in IVLBCL is associated with better outcomes, potentially due to earlier diagnosis and lower systemic disease burden, though caution is warranted when high-risk features coexist. Older age, underlying autoimmune disorders, HPS-associated variant, CNS involvement, nodal involvement, “leukemic” presentation, thrombocytopenia, high LDH levels (>700 IU/L), and dual expression of BCL2 and MYC are instead considered poor prognostic factors [4,12,17,27,35,53].

In conclusion, IVLBCL and other high-risk DLBCL variants require prompt diagnosis and aggressive, individualized systemic therapy. While localized treatments may offer temporary control in frail patients, systemic approaches remain the cornerstone of management. Ongoing advances in molecular diagnostics and targeted therapies hold promise for improving patient outcomes and enabling tailored treatment strategies for these challenging lymphomas.

## 3. IVNKTCL: An Orphan Entity

IVNKTCL is an exceptionally rare malignancy; currently, only **27 cases** with comprehensive immunohistochemical and molecular characterization have been reported in the literature (recently reviewed by Na et al. [57]).

The disease predominantly affects adults, with the **skin** being the most commonly involved site (85% of cases), either as an isolated manifestation or in conjunction with other organs such as CNS, lungs, and kidneys. Consequently, diagnosis is typically established via skin biopsy. High levels of Epstein–Barr virus (EBV) DNA have been detected in peripheral blood, CSF, and pleural effusion samples via polymerase chain reaction (PCR) [57,58].

Histologically, the neoplastic cells are pleomorphic, medium to large in size, exhibiting enlarged nuclei, hyperchromatic nucleoli, and scant cytoplasm. They grow within the vascular lumina in a dys-cohesive pattern. The immunophenotype is consistent with either **T-cell or NK-cell lineage** and shows a cytotoxic profile, with expression of granzyme B and perforin. Immunophenotypic markers such as CD3, CD2, CD7, CD8, and CD56 can be expressed by both T and NK cells, complicating precise lineage determination. However, NK cells are typically negative for CD5, and T-cell receptor (TCR) gene rearrangements are absent. **EBER** is diffusely positive in approximately 85% of cases [57,58,59].

The main differential diagnoses are

ENKTL, a mass-forming lymphoma primarily involving the upper aerodigestive tract;aggressive NK-cell leukemia (ANKL), characterized by younger age of incidence, leukemic involvement of the bone marrow, peripheral blood, liver, and spleen, and an exclusive NK-cell phenotype;EBV-positive nodal T/NK lymphoma, which is primarily nodal in presentation, often of T-cell origin, and rarely demonstrates intravascular growth [59].

IVNKTCL lymphomagenesis is thought to be driven by multiple genetic events occurring in the context of **EBV infection**, particularly affecting genes involved in epigenetic regulation and RNA splicing, that result in a broad disruption of metabolic pathways. Whole-exome sequencing has identified mutations in histone genes, DNA methylation and acetylation enzymes (e.g., *TET2*, *DNMT1*, *HDAC5*), and helicases, as well as copy number losses or truncating mutations in various splicing factors. Conversely, alterations in classical oncogenes and tumor suppressor genes—such as *HRAS*, *MDM2*, *FGFR2*, *VEGFA*, *BCL2L1*, *FAS*, and *AIMP2*—were not evident at the genomic level but were detected via RNA sequencing, suggesting they arise through aberrant alternative splicing mechanisms, as observed in some epithelial neoplasms. Notably, genomic or transcriptomic alterations in adhesion molecules commonly implicated in IVLBCL pathogenesis were absent [39]. Upregulation of the PD-1/PD-L1 axis has also been reported, consistent with other EBV-associated neoplasms [59].

IVNKTCL is a highly aggressive disease. The CHOP chemotherapy regimen appears inadequate, while the addition of ASCT has demonstrated benefit in selected cases; immune checkpoint inhibitors have been claimed as a further potential therapeutic strategy [59]. Despite these interventions, the prognosis remains poor, particularly in patients with multi-organ involvement and/or bone marrow infiltration, with most patients succumbing to the disease within one year [57,58].

## 4. Vasculotropic Systemic Lymphomas: Potential Mimickers of the Original

The diagnosis of intravascular lymphomas requires the careful exclusion of several entities that may mimic their histological or clinical presentation. These mimickers differ in terms of pathogenesis, immunophenotype, distribution, and clinical behavior, and recognizing them is essential to establish an accurate diagnosis and to ensure appropriate therapeutic intervention.

The main differential diagnostic categories include (1) angiotropic/angiocentric lymphomas; (2) systemic, mass-forming lymphomas with secondary intravascular dissemination; and (3) intravascular pseudolymphomas.

**Lymphomas with angiocentric growth patterns** comprise rare B- or T-cell neoplasms that typically consist of a proliferation of large atypical cells with angiotropism, angioinvasion, and angiodestruction. The main entities in this group include lymphomatoid granulomatosis, ENKTL, angioimmunoblastic T-cell lymphoma (AITL), and lymphomatoid papulosis type E (LyP type E) [1]. Lymphomatoid granulomatosis is an EBV-driven B-cell lymphoproliferative disorder with prominent vascular involvement, which typically involves the lungs but may also affect the skin, central nervous system, and kidneys. Histologically, it features a polymorphic infiltrate with variable numbers of EBV-positive large B cells within a reactive T-cell and histiocytic background, often centered around and destroying blood vessels, sometimes with granuloma formation. ENKTL and AITL are T-cell lymphomas typically presenting with mass-forming lesions and/or lymphadenopathy, often associated with an aggressive clinical course. LyP type E, by contrast, is a CD30-positive T-cell lymphoma confined to the skin, characterized by an indolent clinical behavior and a history of recurrent, self-regressing lesions. It does not show selective intravascular growth and usually presents with a polymorphic inflammatory infiltrate, epidermal necrosis, and ulceration. Overall, despite their angiocentric tendencies, these entities generally exhibit extraluminal tumor cell predominance (i.e., growth around but not within vessels), mass formation, and an immunophenotype inconsistent with IVLBCL [1,32]. Intravascular primary effusion lymphoma (intravascular PEL) is another rare entity that may show pure intravascular growth. However, it can be distinguished from IVLBCL by both morphology and immunophenotype. It typically represents a non-germinal center B-cell lymphoma with plasmablastic differentiation, characterized by diffuse positivity for CD138, MUM1, EBER, and HHV-8/LANA1—the latter being essential for diagnosis. These features reflect the distinct cellular origin and biological behavior of intravascular PEL.

Another differential consideration is the **intravascular dissemination of a mass-forming lymphoma**, such as diffuse large B-cell lymphoma (DLBCL) or ALK-negative anaplastic large cell lymphoma. While partial intravascular involvement has been reported in such cases, strict adherence to WHO diagnostic criteria excludes classification as IVLBCL when a dominant mass-forming component is present. Features supporting a diagnosis of IVLBCL include neoplastic proliferation strictly confined to small blood vessels, absence of solid tumor masses, multi-organ involvement, and a consistent B-cell immunophenotype [9,11]. In cases where intravascular proliferation coexists with a mass lesion, histopathological and molecular analyses are necessary to determine whether two distinct lymphomas are present.

**Intravascular pseudolymphomas** are reactive conditions marked by the intravascular accumulation of non-neoplastic, large activated lymphocytes. They are most frequently observed in the skin, often in association with inflammatory dermatoses, trauma, or tumors such as keratoacanthoma. Unlike IVLBCL, these infiltrates are composed of activated T cells—often CD30-positive—and show no systemic involvement [9,30,31].

## 5. Conclusions

Intravascular lymphoma remains a rare and diagnostically challenging entity due to its nonspecific clinical presentation and absence of mass-forming lesions. Our review underscores the importance of maintaining a high index of clinical suspicion and the need for comprehensive histopathological and immunophenotypic evaluation to accurately identify this lymphoma subtype; in fact, it is also crucial to exclude other mimicking conditions, such as lymphomas with intravascular dissemination or angiotropic/angiocentric growth patterns. Emerging evidence of immune evasion mechanisms—such as PD-L1 expression and HLA downregulation—provides further insight into its unique biology. Given the poor prognosis associated with delayed diagnosis, early recognition through a multidisciplinary approach, especially in atypical clinical scenarios, is essential. Further research is warranted to elucidate the molecular basis of the disease, in order to develop more effective diagnostic tools and targeted therapeutic strategies for this rare but aggressive malignancy.

## Figures and Tables

**Figure 1 cancers-17-02355-f001:**
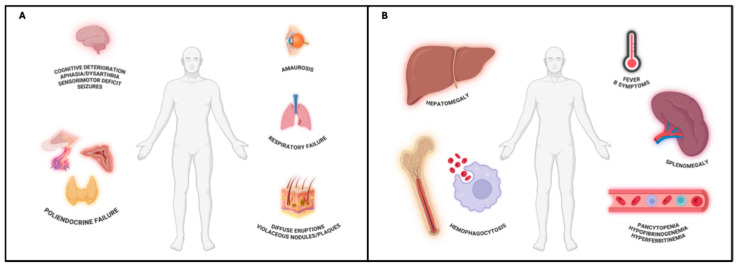
(**A**) IVLBCL, classic variant. Involvement of both skin and visceral organs. CNS manifestations are frequent and may include cognitive deterioration, vertigo, aphasia, sensorimotor deficits, and seizures, often reversible after treatment. Respiratory failure and endocrine dysfunctions (e.g., hypopituitarism, hypothyroidism, adrenal insufficiency) may also occur as both lungs and endocrine system are highly dependent from blood supply in their function. Skin lesions are variable, consisting of violaceous nodules, plaques, or diffuse eruptions. IVLBCL, skin variant, can be considered an early phase of the classical variant, characterized by a better prognosis. Created in BioRender. Crimini, E. 2025. https://BioRender.com/w051ftf (accessed on 17 June 2025) (**B**) IVLBCL, HPS-associated variant. Defined by rapid onset multi-organ failure, pancytopenia, and hepatosplenomegaly due to concurrent hemophagocytic syndrome. Clinical features include persistent fever, B-symptoms, and laboratory evidence of hyperferritinemia, hypertriglyceridemia, and hypofibrinogenemia. The combination of fever, splenomegaly, thrombocytopenia, and elevated ferritin levels may aid early recognition. Created in BioRender. Crimini, E. 2025. https://BioRender.com/r1jemm7 (accessed on 17 June 2025).

**Figure 2 cancers-17-02355-f002:**
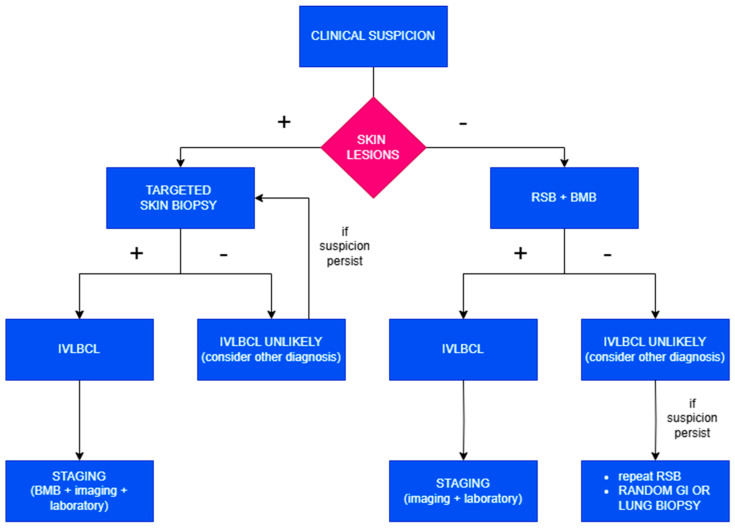
Diagnostic algorithm in case of clinical suspicion of IVLBCL.

**Figure 3 cancers-17-02355-f003:**
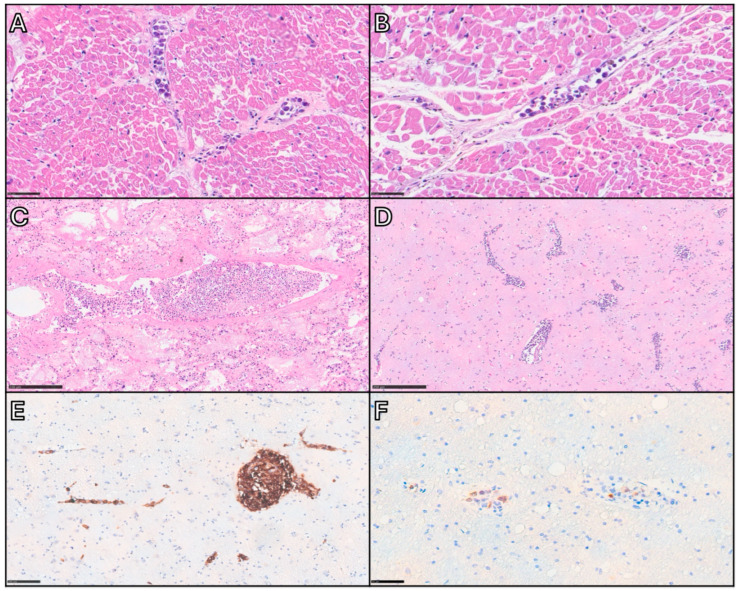
Autopsy case of a middle-aged woman demonstrating key histopathological features of IVLBCL. After a brief clinical course characterized by rapid deterioration, the patient died without an apparent cause. Autopsy was performed, and despite the absence of gross pathological findings, IVLBCL involvement was identified in several vital organs. This case highlights the importance of extensive tissue sampling during autopsy as a critical factor in achieving an accurate postmortem diagnosis. (**A**,**B**) Heart: Small vessels within the myocardium are filled with large, atypical lymphoid cells exhibiting a selective intravascular growth pattern, referable to the so-called cohesive pattern and the marginating pattern, respectively. Scale bar: 50 µm. (**C**) Lung: Neoplastic cells are present within arterioles, venules, and alveolar capillaries. The vein at the center of the section shows a centroluminal growth of neoplastic cells embedded in a fibrin thrombus, with apparent separation from the vascular wall (dys-cohesive pattern). Scale bar: 250 µm. (**D**,**F**) Brain: Intravascular lymphomatous proliferation is visible within small-caliber vessels, with minimal degree of extravascular spread. Immunohistochemistry for CD20 (**E**) and MUM1 (**F**) shows positivity in the atypical cells, confirming their non-germinal center B-cell phenotype. Scale bars: 250, 100, and 50 µm.

**Figure 4 cancers-17-02355-f004:**
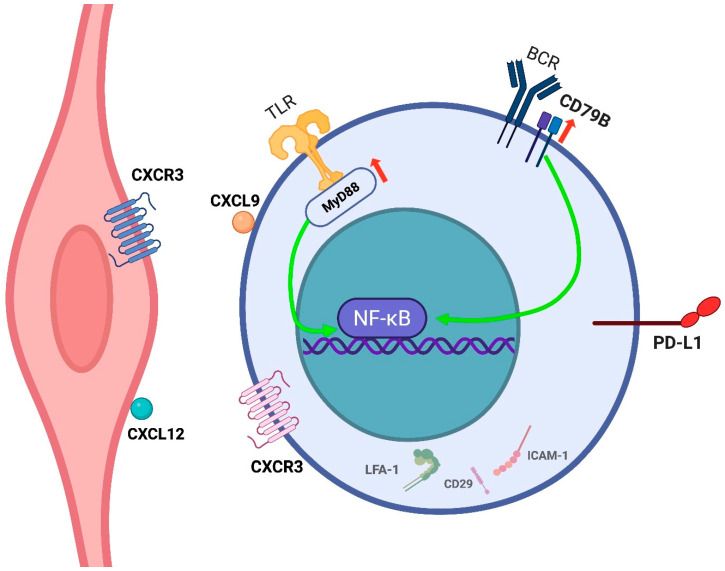
IVLBCL neoplastic cell and its interaction with the endothelium. IVL cells remain confined within the vascular lumen, likely due to defective transendothelial migration resulting from loss of adhesion molecules such as LFA-1 (CD11a/CD18), CD29, and ICAM-1. Chemokine interactions (e.g., CXCL9–CXCR3, CXCL12–CXCR4) may promote endothelial attachment. Overall, IVLBCL cells are able to adhere, to a certain degree, to the endothelium but lack expression of proteins involved in extravasation and parenchymal invasion. Recurrent mutations in MYD88 and CD79B drive NF-κB pathway activation, supporting survival and proliferation. Expression of PD-L1 may contribute to immune evasion within the vascular niche. Created in BioRender. Crimini, E. 2025. https://BioRender.com/vilr6vb (accessed on 17 June 2025).

**Figure 5 cancers-17-02355-f005:**
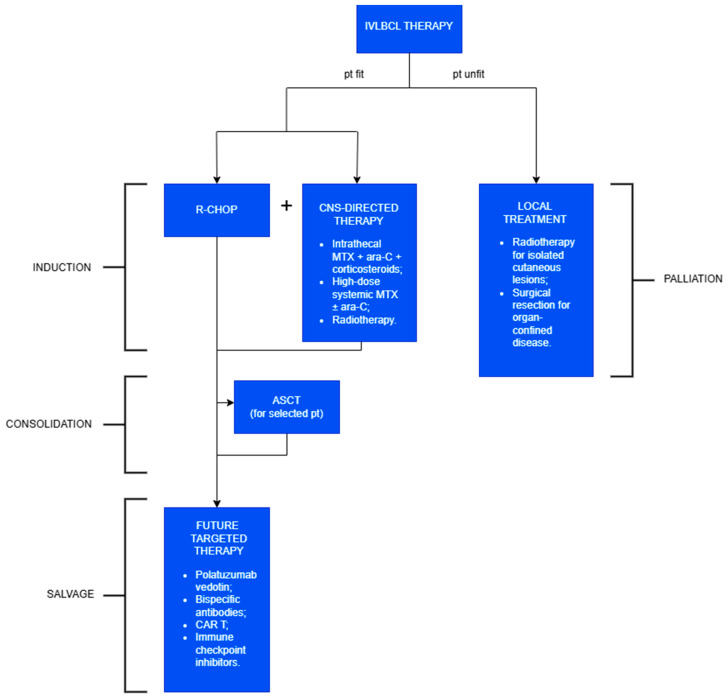
Treatment of IVLBCL.

**Table 1 cancers-17-02355-t001:** This table lists the principal immunohistochemical biomarkers that can be used during the routine diagnostic process, indicating the percentage of cases in which they are expressed and their biological significance.

Biomarker	Expression	Comment
**CD20**	~100%	Pan-B-cell marker; target for rituximab
**CD79a**	~100%	Pan-B-cell marker; compensates for CD20-negativity
**PAX-5**	~100%	B-cell lineage marker (nuclear expression)
**CD10**	11–13%	GCB subtype marker
**BCL6**	26–48%	Suggests GCB subtype when positive in Hans algorithm
**MUM1**	64–95%	Non-GCB subtype marker
**CD5**	22–75%	Aberrant expression, no clear prognostic correlation
**BCL2**	82–91%	Indicates poor prognosis when co-expressed with MYC
**MYC**	50–60%	Indicates aggressive behavior
**PD-L1**	36–44%	Immune evasion and poor prognosis, potential therapeutic target
**CD30**	Rare	Activation marker
**EBER**	Rare	EBV association unclear
**Cyclin-D1**	Negative	Typically associated with mantle cell lymphoma
**CD3**	Negative	Pan-T-cell marker
**ALK**	Negative	Typical of anaplastic lymphomas
**HHV-8/LANA1**	Negative	Typical of intravascular primary effusion lymphoma
**Ki-67**	20–100%	Proliferation index

## Data Availability

Not applicable.

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
