# Peer review of "Intravascular Lymphoma: A Unique Pattern Underlying a Protean Disease"

_cancers, 2025, doi:10.3390/cancers17142355_

Round 1

Reviewer 1 Report

Comments and Suggestions for Authors

The authors present a review of IVL which, however, does not provide novel insights into this disease. 

As pointed out by the authors, IVL is defined as B cell lymphoma according to the WHO Classification. The authors should apply the classification system precisely and stringently throughout their manuscript to clearly distinguish between IVL and rare other lymphoma entities affecting blood vessels. E.g. line 260/261: lymphomas with RS-like morphology have also been included?

Fig. 2 D raises the question whether the case shown represents the border of DLBCL and the infiltration zone of DLBCL into the brain tissue instead of IVL. With the prominent infiltration of the brain parenchyma much extending the blood vessel this case does not show the typical confinement of lymphoma cells to blood vessels characteristic of ICL with only minor and exceptional affection of the brain tissue.

The manuscript might also benefit from further photos illustrating the immunophenotype of the tumor cells and their interaction with endothelial cells.

Author Response

Dear reviewer, thank you for your precious suggestions.

Comment 1: As pointed out by the authors, IVL is defined as B cell lymphoma according to the WHO Classification. The authors should apply the classification system precisely and stringently throughout their manuscript to clearly distinguish between IVL and rare other lymphoma entities affecting blood vessels. E.g. line 260/261: lymphomas with RS-like morphology have also been included?

Answer 1:  The WHO classification serves as the foundation for clinical practice and diagnostic definition of these neoplasms. However, as with any entity—particularly rare ones such as intravascular lymphomas—it is important to acknowledge that exceptions to the model have been reported in the literature. In the histopathological description, it is noted that certain exceptional cases have been described which, although not fully aligned with all WHO criteria, are still best classified as intravascular lymphoma based on the overall features.

Regarding the specific example mentioned (anaplastic or Reed-Sternberg-like cells), the WHO acknowledges that some tumor cells may exhibit anaplastic morphology (WHO Classification of Tumours Editorial Board. Haematolymphoid Tumours. Lyon (France): International Agency for Research on Cancer; 2024. (WHO Classification of Tumours Series, 5th ed.; vol. 11). https://publications.iarc.who.int/637). Furthermore, other studies have reported the presence of RS-like cells (references 11 and 23), which may appear also in various high-grade B-cell lymphomas. These cells are not true Reed-Sternberg cells, as they exhibit a different immunophenotype. We have clarified this point more specifically in lines 284–288.

Comment 2: Fig. 2 D raises the question whether the case shown represents the border of DLBCL and the infiltration zone of DLBCL into the brain tissue instead of IVL. With the prominent infiltration of the brain parenchyma much extending the blood vessel this case does not show the typical confinement of lymphoma cells to blood vessels characteristic of ICL with only minor and exceptional affection of the brain tissue.

Answer 2: The diagnosis was made post-mortem. The possibility that this represented another type of lymphoma with intravascular spread, distinct from IVLBCL, was carefully excluded following a thorough autopsy, which was also correlated with imaging findings. The patient showed no lymphadenopathy or mass-forming solid lesions in any region of the body, including the brain. Therefore, both gross examination and pre-mortem imaging were negative. The observed extravasation was limited exclusively to the brain. Although rare, several reports have described this phenomenon, particularly in advanced or fatal stages of the disease (reference 24). In our case, the degree of extravasation was minimal: it was observed only at high magnification, and neoplastic lymphocytes extended less than 1 mm into the brain parenchyma beyond the vascular lumen. However, more clear pictures are provided in Figure 2.

Comment 3: The manuscript might also benefit from further photos illustrating the immunophenotype of the tumor cells and their interaction with endothelial cells.

Answer 3: We have added additional images illustrating the cytology and immunophenotype of the neoplastic cells, expanding the Figure 2 accordingly.

Reviewer 2 Report

Comments and Suggestions for Authors

This is a very comprehensive and well-organized review of the literature on a rare group of B-cell lymphomas found exclusively intravascularly.  I found the review very helpful and only had a few comments for the authors with the aim of improving the presentation of the manuscript, including:

  1. I would strongly suggest the authors include a high magnification of a tumor lesion so that the readers can see the morphology of the malignant cells within the medium and small vessels.
  2. In addition, regarding Figure 2, I also recommend that for sections B and D, that a higher magnification be included as an insert so that the characteristics of the lesion described in the figure legend can be seen by the reader.  
  3. Also, could the authors add a table showing the key biomarkers for these lymphomas which would be very helpful for the immunodiagnosis of IVL.
  4. I found the cytogenetics and molecular biology sections of the manuscript very helpful.  The authors may mention in these sections that translocation of chromosome 14 may be related to the movement of the Ig Genes behind the strong promoter for the myc or the bcl genes, putative causes of oncogenesis of DHL.  In addition, changes or loss of chromosome 6 such as 6q- could be responsible for HLA loss, a major mechanism of immune evasion to knock out adaptive immunity to tumor.
  5. Related to this last point, the authors need to clarify better the role of HLA-DR in these tumors since their description was rather vague in the text.  These genes are also found on chromosome 6 next to the HLA Class I genes.  Hence, 6q- may also indicate the loss of these genes in those cases showing this chromosomal abnormality.
  6. In the pathogenesis section, I particularly appreciated your discussion regarding the importance of adhesion molecule expression by IVL cells and the potential role of matrix metalloproteinases enzymes for the attachment and sequestration of the lymphoma cells within the small and medium microvessels.  This was an excellent discussion and could inspire other investigators to look at these pathways in these rare tumors.
  7. PD-L1/2 are important genes associated with the pathogenesis of Hodgkin's disease and the over-expression of these checkpoint inhibitors may be the reason Reed-Sternberg cells have been noted in some IVL cases.  You may want to include this point in your discussion of this important marker of IVL.

Finally, the clinical treatment and differential diagnoses sections were well done and sure to be very helpful to clinicians who treat DHL and their variants.

Author Response

Dear Reviewer, thank you for your kind appreciation of our work and for your valuable suggestions on how to improve it.

Comment 1: I would strongly suggest the authors include a high magnification of a tumor lesion so that the readers can see the morphology of the malignant cells within the medium and small vessels.

Comment 2: In addition, regarding Figure 2, I also recommend that for sections B and D, that a higher magnification be included as an insert so that the characteristics of the lesion described in the figure legend can be seen by the reader.  

Answer 1 and 2: We have added additional images illustrating the cytology and immunophenotype of the neoplastic cells, expanding the Figure 2 accordingly.

Comment 3: Also, could the authors add a table showing the key biomarkers for these lymphomas which would be very helpful for the immunodiagnosis of IVL.

Answer 3: We provide to add the requested table, thus illustrating IVLBCL immunophenotype (Table 2).

Comment 4: I found the cytogenetics and molecular biology sections of the manuscript very helpful.  The authors may mention in these sections that translocation of chromosome 14 may be related to the movement of the Ig Genes behind the strong promoter for the myc or the bcl genes, putative causes of oncogenesis of DHL.  In addition, changes or loss of chromosome 6 such as 6q- could be responsible for HLA loss, a major mechanism of immune evasion to knock out adaptive immunity to tumor.

Answer 4: Dear Reviewer, as noted in the manuscript, although translocation events involving chromosome 14 are present, BCL2, BCL6, and MYC are only rarely involved (References 36, 37, 38). Therefore, these events likely have a different significance compared to what is observed in other forms of DLBCL. We clarified this point in lines 352–356. Additionally, regarding chromosome 6 abnormalities, these have been described primarily in the 6q region, and not in 6p, where the HLA genes are located.

Comment 5: Related to this last point, the authors need to clarify better the role of HLA-DR in these tumors since their description was rather vague in the text.  These genes are also found on chromosome 6 next to the HLA Class I genes.  Hence, 6q- may also indicate the loss of these genes in those cases showing this chromosomal abnormality.

Answer 5: HLA downregulation has been demonstrated by the study conducted by Patel et al. (Reference 44) through immunohistochemical analysis. The molecular mechanisms underlying this event in IVLBCL have not been investigated, but only hypothesized. We have further clarified this aspect in lines 438 to 444.

Comment 6: In the pathogenesis section, I particularly appreciated your discussion regarding the importance of adhesion molecule expression by IVL cells and the potential role of matrix metalloproteinases enzymes for the attachment and sequestration of the lymphoma cells within the small and medium microvessels.  This was an excellent discussion and could inspire other investigators to look at these pathways in these rare tumors.

Answer 6: Thank you for your supportive comment. We also believe that this important point warrants further investigation in this direction.

Comment 7: PD-L1/2 are important genes associated with the pathogenesis of Hodgkin's disease and the over-expression of these checkpoint inhibitors may be the reason Reed-Sternberg cells have been noted in some IVL cases.  You may want to include this point in your discussion of this important marker of IVL.

Answer 7: Occasionally, the presence of Reed-Sternberg-like (RS-like) cells has been reported in IVLBCL (References 11 and 23). However, similar cells may also be observed in various high-grade B-cell lymphomas. These are not true Reed-Sternberg cells, as they display a distinct immunophenotype and reflect a completely different etiopathogenesis. Therefore, the term “RS-like” should be interpreted cautiously, as it is often used by pathologists to describe large, somewhat pleomorphic neoplastic lymphoid cells that bear only morphological resemblance to the classic RS cells of Hodgkin lymphoma. We have clarified this point more specifically in lines 284–288.